# Evolution of the Use of Corticosteroids for the Treatment of Hospitalised COVID-19 Patients in Spain between March and November 2020: SEMI-COVID National Registry

**DOI:** 10.3390/jcm10194610

**Published:** 2021-10-08

**Authors:** David Balaz, Philip Erick Wikman-Jorgensen, Vicente Giner Galvañ, Manuel Rubio-Rivas, Borja de Miguel Campo, Mariam Noureddine López, Juan Francisco López Caleya, Ricardo Gómez Huelgas, Paula María Pesqueira Fontán, Manuel Méndez Bailón, Mar Fernández-Garcés, Ana Fernández Cruz, Gema María García García, Nicolás Rhyman, Luis Corral-Gudino, Aquiles Lozano Rodríguez-Mancheño, María Navarro De La Chica, Andrea Torregrosa García, José Nicolás Alcalá, Pablo Díaz Jiménez, Leticia Esther Royo Trallero, Pere Comas Casanova, Jesús Millán Núñez-Cortés, José-Manuel Casas-Rojo

**Affiliations:** 1Department of Internal Medicine, Hospital Universitario San Juan de Alicante, 03550 Alicante, Spain; 2Department of Clinical Medicine, Faculty of Medicine, Miguel Hernández University, 03202 Alicante, Spain; 3Fundación para el Fomento de la Investigación Sanitaria y Biomédica de la Comunitat Valenciana (FISABIO), Conselleria de Sanitat, 46010 Valencia, Spain; 4Department of Internal Medicine, Hospital Universitario de Bellvitge, 08907 Barcelona, Spain; mrubio@bellvitgehospital.cat; 5Department of Internal Medicine, Hospital Universitario 12 de Octubre, 28041 Madrid, Spain; borja.demiguel@gmail.com; 6Department of Internal Medicine, Hospital Costa del Sol, 29603 Málaga, Spain; mariamnoureddine@hotmail.com; 7Department of Internal Medicine, Hospital de Cabueñes, 33394 Gijón, Asturias, Spain; jflcaleya@hotmail.com; 8Department of Internal Medicine, Hospital Regional Universitario de Málaga, 29010 Málaga, Spain; ricardogomezhuelgas@hotmail.com; 9Department of Internal Medicine, Hospital Clínico de Santiago de Compostela, 15706 A Coruña, Spain; paulapesqueira@hotmail.com; 10Department of Internal Medicine, Hospital Clínico San Carlos, 28040 Madrid, Spain; manuelmenba@hotmail.com; 11Department of Internal Medicine, Hospital Universitario Dr. Peset, 46017 Valencia, Spain; mar.fernandez.7823@gmail.com; 12Department of Internal Medicine, Hospital Universitario Puerta de Hierro-Majadahonda, 28222 Madrid, Spain; anafcruz999@gmail.com; 13Department of Internal Medicine, Complejo Hospitalario Universitario de Badajoz, 06010 Badajoz, Spain; geminway21@hotmail.com; 14Department of Internal Medicine, Hospital Moisès Broggi, Sant Joan Despí, 08970 Barcelona, Spain; nicolasrhyman@yahoo.es; 15Department of Internal Medicine, Hospital Universitario Río Hortega, 47012 Valladolid, Spain; lcorral@saludcastillayleon.es; 16Department of Internal Medicine, Hospital Alto Guadalquivir, 23740 Jaén, Spain; aqlozanorod@gmail.com; 17Department of Internal Medicine, Hospital Nuestra Señora del Prado, 45600 Toledo, Spain; maria.navachica@yahoo.com; 18Department of Internal Medicine, Hospital General Universitario de Elda, 03600 Alicante, Spain; andreatg91@gmail.com; 19Department of Internal Medicine, Hospital de Pozoblanco, 14400 Córdoba, Spain; jnalcala58@hotmail.com; 20Department of Internal Medicine, Hospital Universitario Virgen del Rocío, 41013 Sevilla, Spain; pablo_diaz_jimenez@hotmail.com; 21Department of Internal Medicine, Hospital General Defensa, 50009 Zaragoza, Spain; leroyomi@gmail.com; 22Department of Internal Medicine, Hospital Comarcal de Blanes, 17300 Girona, Spain; comascasanova@gmail.com; 23Department of Internal Medicine, Hospital Universitario Gregorio Marañón, 28007 Madrid, Spain; jmillann@med.ucm.es; 24Department of Internal Medicine, Hospital Infanta Cristina University Hospital, 28981 Madrid, Spain; webmaster@fesemi.org

**Keywords:** COVID-19, corticosteroids, Spain, comorbidities

## Abstract

Objectives: Since the results of the RECOVERY trial, WHO recommendations about the use of corticosteroids (CTs) in COVID-19 have changed. The aim of the study is to analyse the evolutive use of CTs in Spain during the pandemic to assess the potential influence of new recommendations. Material and methods: A retrospective, descriptive, and observational study was conducted on adults hospitalised due to COVID-19 in Spain who were included in the SEMI-COVID-19 Registry from March to November 2020. Results: CTs were used in 6053 (36.21%) of the included patients. The patients were older (mean (SD)) (69.6 (14.6) vs. 66.0 (16.8) years; *p* < 0.001), with hypertension (57.0% vs. 47.7%; *p* < 0.001), obesity (26.4% vs. 19.3%; *p* < 0.0001), and multimorbidity prevalence (20.6% vs. 16.1%; *p* < 0.001). These patients had higher values (mean (95% CI)) of C-reactive protein (CRP) (86 (32.7–160) vs. 49.3 (16–109) mg/dL; *p* < 0.001), ferritin (791 (393–1534) vs. 470 (236–996) µg/dL; *p* < 0.001), D dimer (750 (430–1400) vs. 617 (345–1180) µg/dL; *p* < 0.001), and lower Sp0_2_/Fi0_2_ (266 (91.1) vs. 301 (101); *p* < 0.001). Since June 2020, there was an increment in the use of CTs (March vs. September; *p* < 0.001). Overall, 20% did not receive steroids, and 40% received less than 200 mg accumulated prednisone equivalent dose (APED). Severe patients are treated with higher doses. The mortality benefit was observed in patients with oxygen saturation </=90%. Conclusions: Patients with greater comorbidity, severity, and inflammatory markers were those treated with CTs. In severe patients, there is a trend towards the use of higher doses. The mortality benefit was observed in patients with oxygen saturation </=90%.

## 1. Introduction

Effective therapies for patients with coronavirus disease 2019 (COVID-19) are lacking. The role of corticosteroids (CTs) in the management of COVID-19 is of great interest. At the beginning of the pandemic, the use of CTs was contraindicated by several health authorities. The initial recommendation in March and May 2020 of the World Health Organisation (WHO) contraindicated the use of CTs for the management of COVID-19 [1]. Known data from previous studies on MERS, SARS, and influenza pointed to the absence of benefit, increased risk for secondary effects, and some concerns about prolonged virus clearance [2,3,4,5].

Data from the RECOVERY clinical trial published in July 2020 have revealed that low-dose dexamethasone reduced mortality by one-third in ventilated patients with COVID-19 at day 28, compared with patients treated with standard of care alone. Another endpoint was a decrease in mortality by one-fifth in patients receiving oxygen therapy; nonetheless, there was no benefit in patients not requiring respiratory support [6].

The CT beneficial effect is associated with modulation of the inflammatory response. The poor course of the disease is the consequence of an uncontrolled systemic autoinflammatory situation with the virus as the trigger of the so-called “cytokine storm”, an uncontrolled inflammatory response characterised by high levels of IL-1β, IFN-γ, IL-10, and MCP1 that activates the T-helper 1 (Th1) cell response and causes severe lung damage [7,8]. Systemic inflammation takes the form of elevated C-reactive protein, ferritin, procalcitonin, and Il-6, and autopsy results reveal exudative and proliferative changes characteristic of diffuse alveolar damage, including an inflammatory infiltrate comprising macrophages and lymphocytes [9]. Due to these pathogenic mechanisms, prescribing CTs for the treatment of COVID-19 seems reasonable [10,11]. In September 2020, WHO published an updated document [12] recommending the use of CTs in patients with severe and critical COVID-19 infection and contraindicating their use in patients with mild-to-moderate infection. Similarly, the Infectious Diseases Society of America (IDSA) advised the use of CTs, preferring dexamethasone and methylprednisolone or prednisone, in hospitalised patients with severe COVID-19. The indiscriminate and routine use of CT in patients with COVID 19 is not recommended [13] and continues to be discouraged in patients without hypoxemia [14].

Although recent studies showed a beneficial effect in severe and critical cases, the controversy is not over. The main reason is the high heterogeneity of existing studies, making a global analysis and interpretation of the evidence difficult.

Spain has had one of the biggest incidences of infections by SARS-CoV-2 in the world since the outbreak of the pandemic. Although the use of CTs was generally not recommended in the first months of the pandemic, the initial favourable reports from China [15] caused the use of CTs in daily clinical practice from the beginning of the pandemic.

Our aim was to analyse how CTs were used in Spain, using data from the SEMI-COVID-19 Registry. We aimed to identify what type of patients were more likely to receive CTs, the dosage used, how CT use changed over time, and clinical outcomes (mortality).

## 2. Material and Methods

We focused on common comorbidities, biochemical parameters, basic X-ray findings, clinical outcomes, dosage, and timing of corticosteroids in the evolution of pandemics. Blood sampling and basic analysis of chest X-rays were performed on the first day of hospital admission.

### 2.1. The SEMI-COVID-19 Registry

The SEMI-COVID-19 Registry is an ongoing nationwide, multicentre, observational, retrospective cohort registry [16]. A total number of 150 hospitals through the 17 regions of Spain participate in the registry, thus ensuring a representative nationwide sample. All consecutive hospitalised patients with confirmed SARS-CoV-2 infection who were discharged or died were eligible for inclusion. The inclusion criteria were age ≥18 years and first admission to the hospital due to SARS-CoV-2 infection confirmed by reverse transcription–polymerase chain reaction (RT-PCR) testing of a nasopharyngeal sputum sample or bronchoalveolar lavage sample, through a positive result on serological test with a clinically compatible presentation, according to World Health Organisation (WHO) recommendations [17]. The exclusion criteria were subsequent admissions of the same patient and denial or withdrawal of informed consent. The admission and treatment of patients were at the discretion of the attending physicians based on their clinical judgment, local protocols, and the updated recommendations from the Spanish Ministry of Health. Personal data processing strictly complied with the applicable European Union and Spanish laws on biomedical research and personal data protection. The SEMI-COVID-19 Registry has been approved by the Provincial Research Ethics Committee of Málaga (Spain) as per the recommendation of the Spanish Agency of Medicines and Medical Products (AEMPS, for its initials in Spanish). All patients gave informed consent. When there were biosafety concerns and/or when the patient had already been discharged, verbal informed consent was requested and noted on the medical record. The conduct and reporting of the study were carried out pursuant to the STROBE statement guidelines [18].

### 2.2. Procedures

An online electronic data capture system (DCS) was developed for the SEMI-COVID-19 Registry. After receiving training, at least one physician from the internal medicine department in each participating hospital was responsible for acquiring and entering the requested data into the DCS. This work was performed on a voluntary basis, and physicians received no remuneration for it. To ensure the quality of data collection, a database manager and data verification procedures were designed. The study’s scientific steering committee and an independent external agency performed database monitoring. Data analysis and logistics coordination were also carried out by independent external agencies. Alphanumeric sequences of characters were used as identification codes to pseudo-anonymise dissociated patient identifiable data; as such, the DCS did not contain any direct patient identifiers. The database platform is hosted on a secure server, and all information is fully encrypted through a valid TLS certificate.

Multiple variables were retrospectively collected under various headings, which included the following: (1) inclusion criteria, (2) epidemiological data, (3) RT-PCR and SARS-CoV-2 serology data, (4) personal medical and medication history, (5) symptoms and physical examination findings at admission, (6) laboratory (blood gases, metabolic panel, complete blood count, coagulation) and diagnostic imaging tests, (7) additional data at seven days after hospital admission or at admission to the ICU, (8) pharmacological treatment during the hospitalisation and ventilator support, (9) complications during the hospitalisation, and (10) progress after discharge and/or 30 days from diagnosis. The Charlson Comorbidity Index was calculated from the collected data [19].

### 2.3. Statistical Analysis

Quantitative variables are expressed as medians (interquartile range) for not normally distributed variables or means (standard deviation) for normally distributed variables. Categorical variables are expressed as absolute values and percentages. Categorical variables were compared using the chi-square test and quantitative variables using Student’s t-test for variables with a normal distribution and the Mann-Whitney test for non-parametric variables. The alpha significance level was established as 0.05. Analysis of time-dependent variables was carried out by visual inspections of point plots with 95% CI with an overlying smooth curve of conditional means with standard error shadow computed with the LOESS method. A stratified logistic regression was performed to assess the impact of corticosteroids on mortality.

The analysis was conducted with R statistical software.

## 3. Results

In total, 16,717 hospitalised patients due to COVID-19 in Spain were included in the SEMI-COVID-19 registry (Figure 1). In this retrospective cohort study, data from February to November of 2020 were analysed. Corticosteroids were administrated in an important part of hospitalised patients adding to a standard-of-care therapy. A total of 6053 (36.21%) patients received treatment with corticosteroids. We analysed 9595 (57.4%) male and 7122 (42.6%) female patients.

### 3.1. Comorbidities

Among all 16,717 enrolled patients in our cohort, the most frequent comorbidities were: high blood pressure (HBP) *n* = 8542 (51.1%), dyslipidaemia (DLP) *n* = 6612 (39.59%), obesity *n* = 3359 (20.09%), diabetes mellitus with no end organ damage (DMnEOD) *n* = 2416 (14.45%) and atrial fibrillation (AF) *n* = 1840 (11.01%). Asthma was the most common respiratory disease, *n* = 1185 (7.09%). More comorbidities in our cohort are summarised in Table 1.

In addition, we found that the patients with comorbidities including high blood pressure (HBP), dyslipidaemia (DLP), diabetes mellitus no end-organ damage (DMnEOD), diabetes mellitus with end-organ damage (DMEOD), obesity, stable angina, heart failure (HF), chronic obstructive pulmonary disease (COPD), asthma, apnoea hypopnea syndrome (AHS), transient ischemic attack (TIA), dementia, periphery arterial disease (PAD), chronic kidney disease (CKD), leukaemia, lymphoma, connective tissue disease, and AIDS received CTs significantly more often than patients without these comorbidities.

There was no significant difference among the groups according to the presence of depression, anxiety disorder, dementia, stroke, cirrhosis, active neoplasm, and AF.

### 3.2. Biochemical and Gasometrical Parameters

The patients in the CT group had lower oxygen saturation measured by pulse oximetry than the non-CT group at the time of hospital admission (91.3% ± 6.68 vs. 93.8% ± 5.31, *p* < 0.001). Moreover, there were significantly more patients with lower than 93% oxygen saturation at time of admission in the CT group (*n* = 2812 (47.6%) vs. *n* = 2778 (26.8%), *p* < 0.001). In general, comparing basic gasometric parameters between two branches, there were significant differences in pO2, pCO_2_, PaO_2_/FiO_2_.

We observed significant differences between both groups in blood levels of well-known inflammatory markers, such as CRP, procalcitonin, LDH, ferritin, and D-dimer. Complete results are presented in Table 2.

### 3.3. Chest X-ray

The chest X-ray presented bilateral condensations (*n* = 2068 (34.5%) vs. *n* = 2950 (28.1%), *p* < 0.001) or bilateral interstitial infiltrate image (*n* = 3604 (60.1%) vs. *n* = 5285 (50.3%), *p* < 0.001) more often in corticosteroids group. On the contrary, chest X ray without a pathological image (no presence of bilateral condensation or interstitial infiltrate) was more frequent in the non-CT group (*n* = 3001 (51.1%) vs. *n* = 5679 (54%), *p* < 0.001) and (*n* = 1877 (31.3%) vs. *n* = 4038 (38.4%), *p* < 0.001), respectively. No difference was observed between both groups regarding the presence of pleural effusion (Table 2).

### 3.4. Administration of Corticosteroids during Pandemic

We analysed the change in the use of CT over time. When all patients hospitalised were analysed, we observed that in the month of March, 33.7% received steroids, while in September, 76.4% of patients received corticosteroids (*p* < 0.001, Figure 2A). When the analysis was restricted to severe patients, oxygen saturation <=90%, according to WHO criteria [1] the proportion of patients receiving corticosteroids increased from 51.6% in March month to 85.7% in September (*p* = 0.02, Figure 2B).

Accumulated prednisone equivalent dose (APED) used changed from March when 6.2, 5.7, 3.7 and 8.8% of patients received 200–400, 400–600, 600–800 or more than 800 mg, respectively, to September, when 31.4, 11.2, 13.4 and 6.7% received those doses (*p* < 0.001, Figure 3A). When the analysis was restricted to patients with severe pneumonia, the rise increased from 8.5, 8.1, 5.7, 15.3% to 14.2, 21.4, 7.1, 35.7, 21.4% (*p* < 0.001, Figure 3B).

When we analysed patients’ outcomes, higher mortality is noticed in patients receiving CT (27.9% vs. 15.7% *p* < 0.001). When the analysis was restricted to severe patients, mortality was lower in the group of patients treated with steroids at 40.7% (CI95%, 38.6–42.9%) vs. no steroids at 46.2% (CI95%, 43.91–48.6%) (*p* < 0.001). Mortality varied with steroid dose across oxygen saturation groups (Figure 4). To evaluate this finding, a logistic regression model of mortality risk factors was fit for each group of oxygen saturation. For patients with an oxygen saturation on admission above 94%, corticosteroids were associated with higher mortality across all dose ranges (OR 2.1, 1.7, 1.9, 2.0, *p* < 0.05 for all categories). In patients with an oxygen saturation between 91 and 94%, they were not associated with a higher nor a lower risk (OR 1.2, 1.3, 0.9, 0.9, *p* > 0.05 for all categories). In severe patients, with an oxygen saturation of 90% or less, corticosteroids were associated with a lower mortality risk at doses higher than 400 mg of APED (OR 1 (*p* = 0.7), 0.7 (*p* = 0.007), 0.5 (*p* = 0.0006), 0.6 (*p* = 0.0002), Table 3) in comparison to the group that did not receive CTs.

## 4. Discussion

Our retrospective analysis showed that CTs were used mostly in severe and critical cases of COVID-19. Patients that received CTs were generally older, more frequently male, and more affected by comorbidities. The prevalence of hypertension, cardiovascular diseases, and diabetes has been reported to be two- to threefold higher in patients with a severe type of COVID-19 than in their non-severe counterparts, and these comorbidities have been described as risk factors for mortality [20,21,22,23].

The most common comorbidity in our cohort was high blood pressure with a prevalence of 51.1% which is higher than in other published data. The prevalence of high blood pressure in other works ranges between 10 and 34% [24]. Higher prevalence in our cohort is congruent with national data of arterial hypertension in Spain, where one in three adults is hypertensive (66% in those >60 years) [25].

Patients receiving CTs were more affected by comorbidities but also presented with worse baseline biochemical, gasometric, and radiological parameters at the time of admission in comparison with the non-CT group.

Oxygen levels were lower in the group receiving CT. Additionally, other analytical markers associated with severe COVID [26] such as CRP, procalcitonin, LDH, creatinine, urea, ferritin, and D-dimer, were higher in the group of patients receiving CT. Moreover, X-ray findings were also worse in the group receiving CT.

It is noteworthy that two well-known biomarkers, lymphopenia and thrombocytopenia, did not reach a significant difference between the two analysed groups. This could be due to a point data collection at hospital admission, as lymphopenia and thrombocytopenia usually develop during the course of infection and hospitalisation.

The use of CTs changed dramatically after the month of June 2020. That month, a press release revealed data from the RECOVERY trial [6]. In September, 80% of hospitalised COVID-19 patients received CTs, whereas only 35% received them in March. The dose of CT used did also change over time. The initial daily dose of 1 mg/Kg of ideal body weight was associated with the highest mortality reduction in RCTs of non-viral ARDS and large observational studies in SARS-CoV-2 and H1N1 pneumonia [27]. A Spanish semi-randomised study investigated methylprednisolone (3 days each, 80 mg and 40 mg, respectively) in 85 COVID-19 (56 CST, 29 control) hypoxemic patients. CT was associated with a reduced risk of admission to ICU, non-invasive ventilation, or death. [28] The RECOVERY trial showed significant mortality decreases with 6 mg of dexamethasone once daily for up to 10 days.

In the light of those results, the WHO published new guidance regarding the use of corticosteroids. There is little doubt as to whether corticosteroids are effective for the treatment of critically ill patients with COVID-19, as well as for severe patients. The WHO recommendations do recommend corticosteroids only for these subgroups. Severe COVID-19 was defined as those with an oxygen saturation equal to or below 90%. However, caution was advised, as 90% was admitted to being an arbitrary threshold. We, therefore, performed a stratified analysis and multivariate analysis to determine the optimal oxygen saturation threshold at which CTs seemed to be useful and at what dose. In a previous communication from our centre, we reported better outcomes with doses slightly higher than those used in the RECOVERY trial. [29]

This study found that corticosteroids were associated with higher mortality in patients with an oxygen saturation above 94%. This is in accordance with the results of the RECOVERY trial, in which a possibility of harm was reported among patients that did not need oxygen support [29]. On the other hand, we cannot rule out that some other factor may be contributing to this finding that was not taken into consideration in the multivariate analysis. Therefore, we would approach this finding with caution. CTs appeared to be neutral when saturation was between 91% and 94%, and clearly beneficial when used in patients with an oxygen saturation of 90% or less.

An interesting finding is that, again, the dose found to be effective is a dose higher than the dose of the RECOVERY trial, as the minimum effective dose was an APED of 400–600 mg. The dose used in the RECOVERY trial was an APED of 380 mg. Moreover, there seems to be a trend for more effectiveness at higher doses (Figure 4), raising the question of whether high dose pulses of steroids (>500 mg of prednisone/day or similar) would be even more effective.

Our study has several limitations. This is an observational, retrospective study in which data were collected by a large team of researchers, which could have led to heterogeneity in data entry and validation. The study was not designed for investigating the effectiveness of corticosteroids or other clinical outcomes related to their use, such as adverse effects, days of hospital admission, or mortality. In our study, the baseline characteristics between the CT use group and the non-CT use group were unbalanced.

On the other hand, our study has several strengths. The first is the sample size, and the second one is that the study population spans the entire geographical area of the healthcare system. Another advantage is that the conclusions have been drawn from the daily clinical activity, and therefore, suggestions for improving COVID-19 management can be used in a real-world setting.

In conclusion, we can affirm that from February to November 2020, corticosteroid use in Spanish hospitals was more common in patients with worse biochemical, gasometric, and radiological status on hospital admission. This group also presented with more comorbidities. The administration of corticosteroids increased dramatically after the publication of the RECOVERY trial, showing effectiveness in hospitalised patients with COVID-19 needing oxygen support. Patients with oxygen saturation equal to or below 90% had better outcomes if CTs were administered, especially a dose above 400 mg APED. In patients with oxygen saturation above 90%, they were either neutral or harmful.

## Figures and Tables

**Figure 1 jcm-10-04610-f001:**
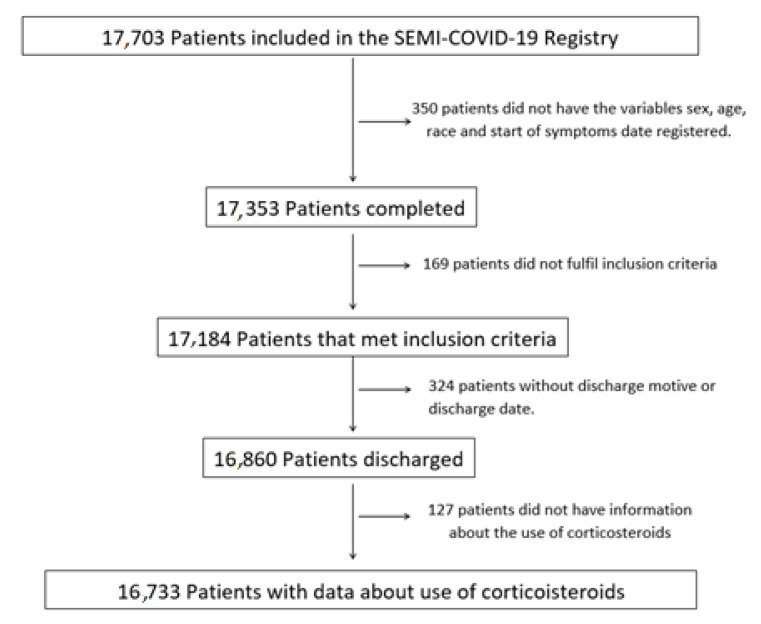
Patient selection flowchart.

**Figure 2 jcm-10-04610-f002:**
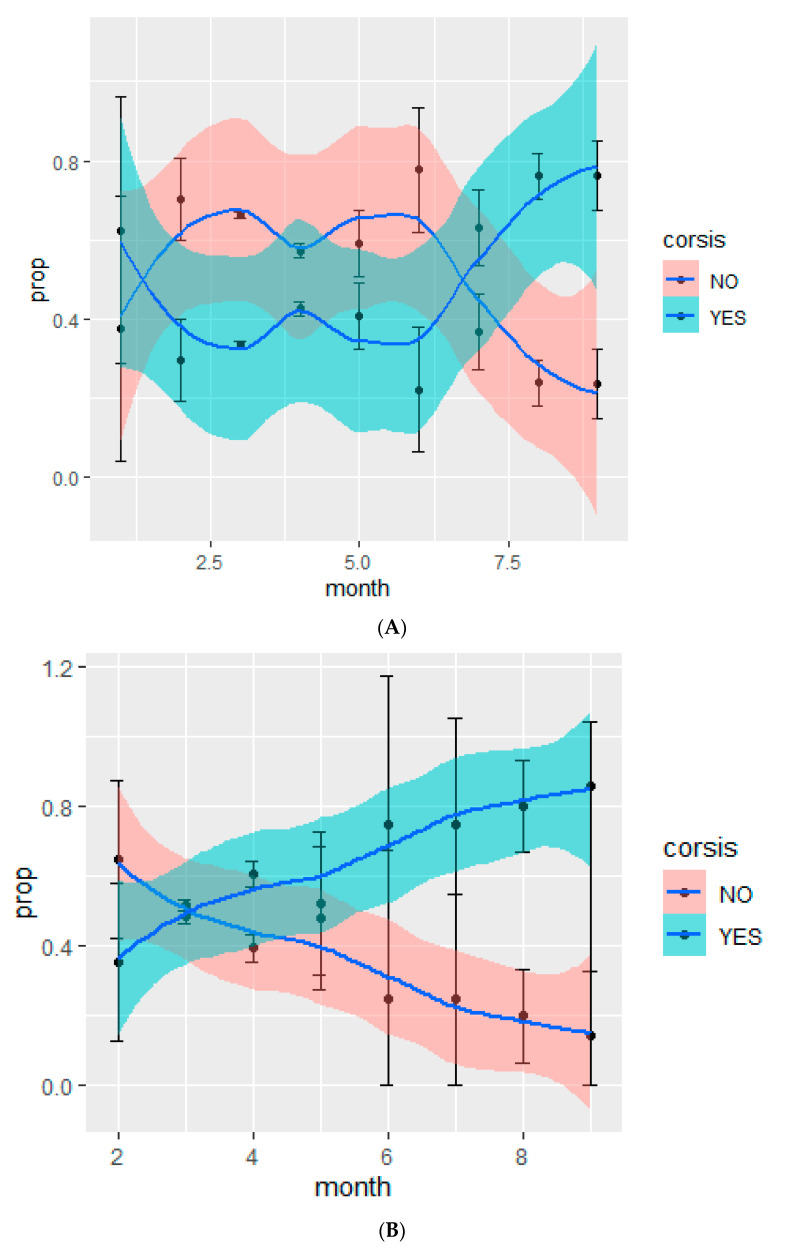
(**A**) Proportion of patients receiving and not receiving corticosteroids during the pandemic in hospitalised patients with COVID-19; (**B**) proportion of patients receiving and not receiving corticosteroids during the pandemic in patients with severe COVID-19 (modified WHO criteria). Prop: proportion, corsis: systemic corticosteroids.

**Figure 3 jcm-10-04610-f003:**
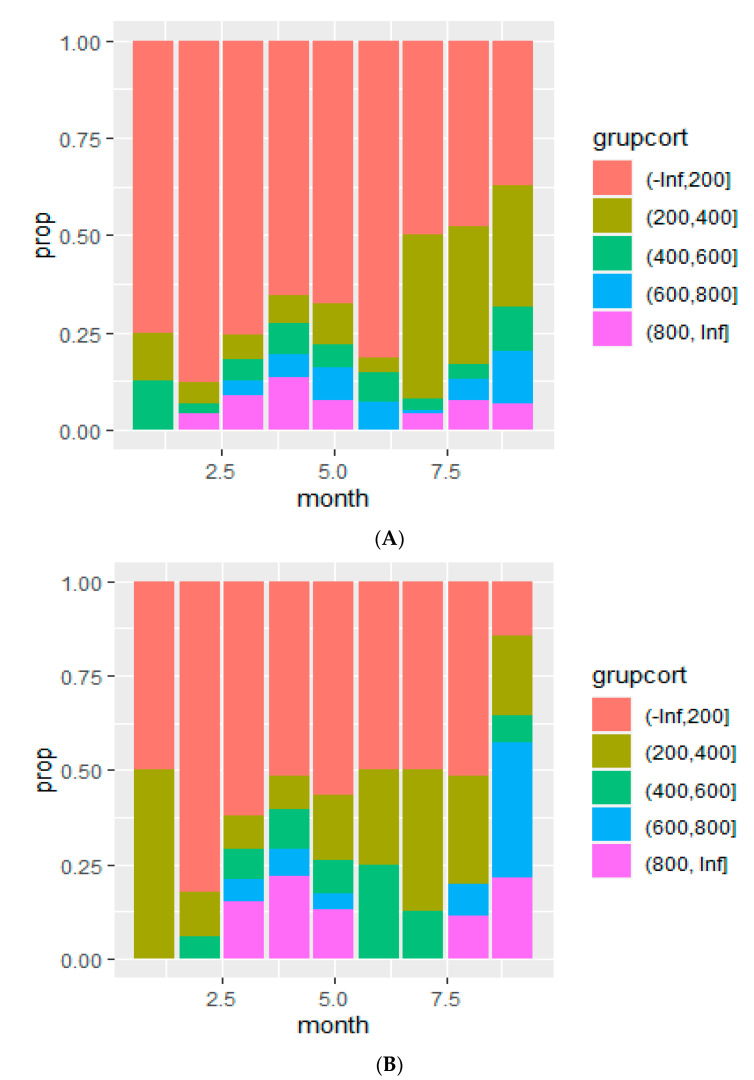
(**A**) Evolution of the dose of accumulated prednisone equivalents administered to patients admitted with COVID-19; (**B**) evolution of the dose of accumulated prednisone equivalent administered dose to patients admitted with severe COVID-19 (modified WHO criteria).

**Figure 4 jcm-10-04610-f004:**
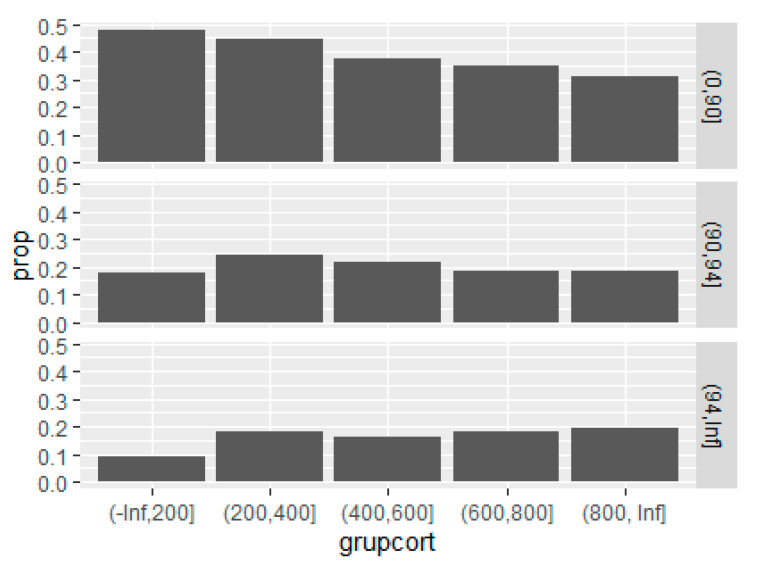
Mortality of patients by corticosteroid dose (accumulated prednisone equivalent dose) and oxygen saturation group.

**Table 1 jcm-10-04610-t001:** Basic characteristics and comorbidities in the cohort.

Variable	Total Cohort	CT	nCT	*p*-Value
Participants, *n* (%)	16,717	6053 (36.21)	10,664 (63.79)	<0.001
Age (years), Mean (SD)	67.33 (16.13)	69.65 (14.58)	66.01 (16.80)	<0.001
Men, *n* (%)	9595 (57.39)	3840 (63.44)	5755 (54.07)	<0.001
HBP, *n* (%)	8542 (51.1)	3450 (57)	5092 (47.75)	<0.001
DMnEOD, *n* (%)	2416 (14.45)	983 (16.24)	1433 (13.44)	<0.001
DMEOD, *n* (%)	896 (5.36)	363 (6)	533 (5)	0.0071
DLP, *n* (%)	6612 (39.55)	2601 (43.02)	4011 (37.64)	<0.001
Alcohol abuse, *n* (%)	751 (4.62)	313 (5.35)	438 (4.22)	<0.001
Smoking, *n* (%)				
Never	11,137 (69.49)	3765 (65.12)	533 (5.20)	
Ex-smoker	4053 (25.29)	1714 (29.64)	2339 (22.83)	<0.001
Smoker	835 (5.2)	302 (5.22)	7372 (71.96)	
AF, *n* (%)	1840 (11.01)	711 (11.76)	1129 (10.59)	0.022
Anxiety disorder, *n* (%)	1292 (7.74)	486 (8.05)	806 (7.57)	0.273
Depression, *n* (%)	1757 (10.5)	669 (11.09)	1083 (10.17)	0.069
Obesity, *n* (%)	3359 (20.09)	1463 (26.37)	1896 (19.33)	<0.001
Neurodegenerative disease, *n* (%)	1484 (8.88)	540 (8.92)	944 (8.85)	0.899
Angina, *n* (%)	591 (3.53)	258 (4.26)	333 (3.12)	0.001
HF, *n* (%)	1187 (7.1)	491 (8.11)	696 (6.53)	0.001
COPD, *n* (%)	1132 (6.77)	585 (9.67)	547 (5.13)	<0.001
Asthma, *n* (%)	1185 (7.09)	474 (7.8)	711 (6.67)	0.006
TIA, *n* (%)	789 (4.72)	322 (5.32)	467 (4.38)	0.007
Stroke, *n* (%)	462 (2.76)	177 (2.92)	285 (2.67)	0.372
Hemiplegia, *n* (%)	272 (1.62)	165 (1.54)	107 (1.76)	0.310
Dementia, *n* (%)	1659 (9.9)	559 (9.24)	1100 (10.3)	0.03
PAD, *n* (%)	765 (4.58)	315 (5.21)	450 (4.23)	0.004
Cirrhosis, *n* (%)	175 (1.04)	72 (1.19)	103 (0.97)	0.19
CKD, *n* (%)	1008 (6.18)	440 (7.28)	568 (5.33)	<0.001
Active Neoplasm, *n* (%)	1034 (6.18)	405 (6.69)	629 (5.9)	0.05
Leukaemia, *n* (%)	194 (1.1)	96 (1.59)	98 (0.92)	<0.001
Lymphoma *n* (%)	220 (1.3)	95 (1.57)	125 (1.17)	0.04
CTD, *n* (%)	416 (2.49)	204 (3.37)	212 (1.99)	<0.001
Lupus, *n* (%)	31 (0.1)	19 (0.31)	12 (0.11)	<0.001
Rheumatoid arthritis, *n* (%)	240 (1.43)	118 (1.95)	128 (1.2)	<0.001
AIDS, *n* (%)	51 (3.05)	28 (0.46)	23 (0.22)	0.009
AHS, *n* (%)	575 (6.16)	451 (7.5)	555 (5.2)	<0.001
CCI > 3, *n* (%)	2899 (17.73)	1225 (20.6)	1674 (16.1)	<0.001

HBP: high blood pressure; DMnEOD: diabetes mellitus no end-organ damage; DMEOD: diabetes mellitus end-organ damage; DLP: dyslipidaemia; AF: atrial fibrillation; HF: heart failure; COPD: chronic obstructive pulmonary disease; TIA: transient ischemic attack; PAD: peripheral artery disease; CKD: chronic kidney disease; CTD: connective tissue disease; AHS: apnoea hypopnea syndrome; CCI: Charlson Comorbidity Index; AIDS: acquired immune deficiency syndrome

**Table 2 jcm-10-04610-t002:** Analytical, gasometric, and X-ray characteristics of the CT and non-CT group.

Variable	Total Cohort	Corticosteroids	Not Corticosteroids	*p*-Value
O_2_ saturation %, SD	92.9 ± 5.9	91.3 ± 6.68	93.8 ± 5.31	<0.001
O_2_ sat < 93% *n* (%)	5590 (34.3)	2812 (47.6)	2778 (26.8)	<0.001
PO_2_ mmHg	68.82 ± 21.8	66.2 ± 21.4	70.9 ± 21.9	<0.001
PCO_2_ mmHg	35.52 ± 8.68	35.3 ± 8.8	35.7 ± 8.59	0.01
pH	7.43 ± 0.18	7.44 ± 0.14	7.43 ± 0.22	0.03
PaO_2_/FiO_2_	282.82 ± 98.29	266 ± 91.1	301 ± 101	<0.001
Leucocytes 10^6^/L	6300 (4800–8600)	6700 (4978–9250)	6110 (4700–8160)	<0.001
Lymphocytes 10^6^/L	940 (690–1300)	860 (600–1200)	1000 (700–1360)	0.14
Neutrophils 10^6^/L	4600 (3220–6720)	5100 (3500–7552)	4340 (3100–6292)	<0.001
Platelets 10^9^/L	190 (148–247)	187 (146–246)	192 (149–247)	0.078
Haemoglobin gr/dL	13.8 (12.6–15)	13.6 ± 1.96	13.7 ± 1.85	0.047
CRP (C-reactive protein mg/dL)	61.5 (20.2–129.79)	86 (32.7–160)	49.3 (16–109)	<0.001
Creatinine (mg/dL)	0.9 (0.73–1.16)	0.97 (0.78–1.25)	0.87 (0.71–1.1)	<0.001
Urea (mg/dL)	37 (27–55)	42 (30–62)	35 (26–51)	<0.001
LDH (lactate dehydrogenase UI/L)	321 (247–431)	353 (268–468)	305 (237–411)	<0.001
Ferritin (µg/mL)	608 (285–1230.8)	791 (393–1534)	470 (236–996)	<0.001
Procalcitonin (ng/mL)	0.1 (0.05–0.22)	0.13 (0.07–0.3)	0.09 (0.05–0.18)	<0.001
Dimer-D (ng/mL)	669 (373–1261)	750 (430–1400)	617 (345–1180)	<0.001
X-ray consolidation				
No	8687 (52.6)	3001 (50.1)	5679 (54)	<0.001
Unilateral	2796 (16.9)	920 (15.4)	1881 (17.9)	
Bilateral	5016 (30.4)	2068 (34.5)	2950 (28.1)	
X-ray interstitial infiltrates/ground glass opacities				
No	5915 (35.84)	1877 (31.3)	4038 (38.4)	<0.001
Unilateral	1699 (10.29)	515 (8.59)	1184 (11.3)	
Bilateral	8889 (53.86)	3604 (60.1)	5285 (50.3)	
Pleural effusion				
No	15,727 (95.32)	5712 (95.3)	10,015 (95.4)	0.58
Unilateral	497 (3.01)	189 (3.15)	308 (2.93)	
Bilateral	274 (1.6)	94 (1.57)	180 (1.71)	
Outcome				
Discharge home	12,119 (72.42)	3946 (65.12)	8173 (76.6)	<0.001
Discharge care centre	1226 (7.32)	422 (6.96)	804 (7.55)	
Death	3388 (20.24)	1692 (27.92)	1696 (15.9)	

**Table 3 jcm-10-04610-t003:** Multivariate analysis with a logistic regression model by oxygen saturation group.

Variable	Oxygen Saturation above 94		Oxygen Saturation between 94–91%		Oxygen Saturation Less or Equal to 90%	
	OR (95% CI)	*p*-Value	OR (95% CI)	*p*-Value	OR (95% CI)	*p*-Value
APED (accumulated prednisone equivalent dose) 200–400	2.1 (1.5–2.9)	<0.001	1.2 (0.9–1.7)	0.19	1 (0.8–1.4)	0.7
APED 400–600	1.7 (1.2–2.4)	0.003	1.3 (0.9–1.8)	0.12	0.7 (0.5–0.9)	0.007
APED 600–800	1.9 (1.3–2.8)	0.001	0.88 (0.6–1.4)	0.59	0.5 (0.4–0.8)	0.0006
APED > 800	2.0 (1.4–2.7)	<0.001	0.9 (0.7–1.3)	0.66	0.6 (0.5–0.8)	0.0002
Charlson index >= 3	2.4	<0.001	2.1 (1.7–2.6)	<0.001	1.9 (1.6-2.3))	<0.001
Lymphocytes > 1000 cells/mL	0.6 (0.5-0.7)	<0.001	0.6 (0.5–0.8)	<0.001	0.7 (0.6–0.8))	<0.001
Age 60–70 y	2.7 (1.8–4.1)	<0.001	2.2 (1.5–3.5)	<0.001	2.4 (1.7–3.4)	<0.001
Age 70–80 y	7.3 (5.2–10.6)	<0.001	8.4 (5.9–12.3)	<0.001	6.0 (4.4–8.2)	<0.001
Age >80 y	23.7 (16.9–33.9)	<0.001	18.6 (13.1–27.2)	<0.001	13.9 (10.3–19.2)	<0.001
LDH (lacate dehydrogenase) 250–500 IU/L	1.5 (1.2–1.9)	0.0005	1.66 (1.3–4.1)	<0.001	2.5 (1.9–3.2)	<0.001
LDH > 500 IU/L	3.8 (2.9–5.1)	<0.001	3.1 (2.3–4.1)	<0.001	6.5 (4.9–8.7)	<0.001
Woman	0.8 (0.7–1.0)	0.13	0.7 (0.6–0.9)	0.0006	0.7 (0.6–0.8)	<0.001
Tocilizumab	2.5 (1.7–3.5)	<0.001	2.7 (2.0–3.6)	<0.001	1.09 (0.8–1.8)	0.47
Remdesivir	0.4 (0.1–1.0)	0.08	0.5 (0.2–1.0)	0.069	0.8 (0.3–1.8)	0.6

## Data Availability

The conduct and reporting of the study were carried out pursuant to the STROBE statement guidelines. An online electronic data capture system (DCS) was developed for the SEMI-COVID-19 Registry. Data analysis and logistics coordination were also carried out by independent external agencies. Alphanumeric sequences of characters were used as identification codes to pseudo-anonymise dissociated patient identifiable data; as such, the DCS did not contain any direct patient identifiers. The database platform is hosted on a secure server, and all information is fully encrypted through a valid TLS certificate.

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
