# Peer review of "Evolution of the Use of Corticosteroids for the Treatment of Hospitalised COVID-19 Patients in Spain between March and November 2020: SEMI-COVID National Registry"

_jcm, 2021, doi:10.3390/jcm10194610_

Round 1

Reviewer 1 Report

A well-designed study representing local implementation of WHO COVID guidelines in Spain. The results a properly analyzed and nicely graphically presented, discussed with relation to clinical and pathophysiological aspects. The current study is in line with the current international research data (incl. our own local data on implemenation of corticosteroid use in COVID).

Author Response

We thank the reviewer for the very positive comments. It is nice to see that no questions arised during the evaluation.

Reviewer 2 Report

Although this study is a retrospective study, it is a large-scale, multicenter study that is very interesting and meaningful. There is not much-established evidence for the administration of corticosteroids in COVID-19, and the selection of patients who are likely to benefit and the optimal treatment regimen have not yet been established. I think this research can be one of the pieces of evidence. Although a sub-analysis shows that administration of corticosteroids to patients with oxygen saturation above 94 clearly increases mortality, I think this is not very consistent with the results of clinical practice. Is it possible that this group has some background of having to receive corticosteroids in the absence of respiratory failure and has a higher mortality rate than other well prognostic groups who did not receive corticosteroids? Please mention this point in the discussion.

Author Response

We thank the reviewer for the very insightful comment. 

Regarding the increase in mortality found in the subgroup with O2sat>94% this finding is in accordance with the results of the RECOVERY trial. Where patients that did not need supplemental O2 had increased mortality with corticosteroids although not significant. And the harmful effect was considered only a possibility. As a measure of comorbidities, we adjusted in the multivariable analysis by the Charlson index, so we feel it unlikely that there would be some other condition explaining the results. On the other hand, it cannot be ruled out that some other factor is influencing the results as this is an observational study with its limitations.

Nevertheless, we have tried to clarify this point in the discussion. 

Round 2

Reviewer 2 Report

You responded exactly to what I have pointed out.
I think this paper is worth publishing.